# Lignin Synthesis, Affected by Sucrose in Lotus (*Nelumbo nucifera*) Seedlings, Was Involved in Regulation of Root Formation in the *Arabidopsis thanliana*

**DOI:** 10.3390/ijms23042250

**Published:** 2022-02-18

**Authors:** Libao Cheng, Chen Zhao, Minrong Zhao, Yuyan Han, Shuyan Li

**Affiliations:** 1School of Horticulture and Plant Protection, Yangzhou University, Yangzhou 225009, China; zhaochen2878359757@163.com (C.Z.); wayzyj@163.com (M.Z.); Hyyluckystar@163.com (Y.H.); 2College of Guangling, Yangzhou University, Yangzhou 225009, China; lsydbnd@163.com

**Keywords:** lotus, adventitious roots, lignin, sucrose, *NnLAC17*

## Abstract

Adventitious roots (ARs) have an unmatched status in plant growth and metabolism due to the degeneration of primary roots in lotuses. In the present study, we sought to assess the effect of sucrose on ARs formation and observed that lignin synthesis was involved in ARs development. We found that the lignification degree of the ARs primordium was weaker in plants treated with 20 g/L sucrose than in 50 g/L sucrose treatment and control plants. The contents of lignin were lower in plants treated with 20 g/L sucrose and higher in plants treated with 50 g/L sucrose. The precursors of monomer lignin, including p-coumaric acid, caffeate, sinapinal aldehyde, and ferulic acid, were lower in the GL50 library than in the GL20 library. Further analysis revealed that the gene expression of these four metabolites had no novel difference in the GL50/GL20 libraries. However, a laccase17 gene (*NnLAC17*), involved in polymer lignin synthesis, had a higher expression in the GL50 library than in the GL20 library. Therefore, *NnLAC17* was cloned and the overexpression of *NnLAC17* was found to directly result in a decrease in the root number in transgenic *Arabidopsis* plants. These findings suggest that lignin synthesis is probably involved in ARs formation in lotus seedlings.

## 1. Introduction

The lotus is an aquatic plant that contains multiple nutrients. The lotus is believed to be an important vegetable and is wildly cultivated in the Yangtze River and Yellow River basins owing to the suitable climate [1,2]. Adventitious roots (ARs) are secondary roots that play an important role in the water and nutrient uptake in plants, especially in lotuses, which have underdeveloped principal roots. ARs are always located on the hypocotyl of seedlings in lotuses, and approximately two weeks are required for AR emergence [3]. There are three developmental stages in the entire period of AR formation [4,5]: the first stage is the induced period, where the normal cell is divided and developed into meristematic cells; the second stage is the development stage, where the root primordium is formed from meristematic cells [6]; and the third stage is where the root primordium emerges from the epidermis of the stem or leaf [7,8]. The signaling transduction pathway of plant hormones is found to be involved in various metabolic processes of plant growth. In the past years, indole-3-acetic acid (IAA) has been reported to play an important role in root formation at the inducted, developmental, and emergence stages [9,10]. At the same time, ethylene is another important factor to regulate ARs formation, and the role of ethylene in AR formation is mainly carried out at the induction stage [11,12,13]. IAA and ethylene can also cooperatively regulate AR development. The metabolism and sensitivity of auxin are affected by ethylene. However, the role of ethylene is also regulated by IAA [14,15,16]. Sugar is involved in root development [17]. In fact, the role of sucrose in the regulation of root formation has been revealed to be similar to that of some types of plant hormones. Further, sucrose can promote the formation of root primordium and late development [10,18]. In lotuses, it is found that there is an interaction between sucrose and IAA during AR formation. The exogenous application of sucrose can affect the IAA content, and exogenous IAA reversely changes the endogenous sucrose content during AR formation [19]. Lignin is an important component of cells during plant growth and metabolism [20]. Normally, lignin, which is a polyphenolic polymer, contains three types of monomers, namely the S-type, H-type, and G-type [21]. According to recently reported data, a close relationship exists between lignin metabolism and AR formation. Cho et al. [22] reveal that indolebutyric acid (IBA) treatment promotes root primordial development, but suppresses lignin synthesis. Peroxidase genes are responsible for lignin synthesis, and this peroxidase gene expression is found to be upregulated after IBA treatment [23,24]. Moreover, two different types of *Ebenus cretica* (rooting and non-rooting genotypes) are found to have different lignin and peroxidase activities [25]. In sweet potato, the formation of the storage root is reduced and vascular lignification is promoted after the exogenous application of plant hormones [26]. A high expression of genes related to lignin synthesis occurs in wild species (not from storage roots) relative to cultivated species [27]. In mung bean seedlings, the genes responsible for lignin biosynthesis are down-regulated after IBA treatment, which significantly promotes AR formation [28]. The development of rice roots is previously found to be inhibited, whereas lignin deposition is increased in cell walls under conditions of stress [29].

Sucrose is an important signal molecular that is involved in various biological process in plant. In the present study, we found that sucrose was involved in the lignin synthesis of lotus seedlings, and the lignin content, which was regulated by *NnLAC17*, affected root formation based on the changes in physiological and biochemical indexes, microstructure observation, RNA-seq (RNA-sequencing) technique, and transgenic plant analysis. This finding provided evidence that ARs formation is regulated by multiple factors, and, at the same time, it also gave new insight on the possible regulatory network of the ARs formation of lotus seedlings.

## 2. Results

### 2.1. Effect of Sucrose on AR Formation

It was shown that seedlings treated with 20 g/L sucrose had a higher number of ARs than that of control plants after transferal to water. Such findings indicated that 20 g/L sucrose significantly promoted the formation of ARs. The developmental process of ARs was markedly delayed in seedlings treated with 50 g/L sucrose. In fact, the seedlings cultured on the MS culture medium containing 20% sucrose had a more adventitious root number than that of 50% sucrose (Figure 1a). Further analysis revealed that the number of ARs was lower in plants treated with 50 g/L sucrose than in control plants and 20 g/L sucrose treatment (Figure 1b). In summary, we concluded that low concentrations of sucrose accelerated ARs formation, whereas high concentrations inhibited this process.

### 2.2. Observation of the Paraffin Sections

The microstructure of hypocotyls was analyzed for ARs development following treatment with 20 mg/L and 50 mg/L sucrose. The primordia of ARs were formed around the stomata in the hypocotyl. The induction and development of ARs took approximately 2 days after treatment with 20 g/L sucrose; however, a longer time was taken by treatment with 50 g/L sucrose. In the last stage (expressed stage) of ARs development, a greater number of ARs broke through the epidermis of hypocotyl following treatment with 20 mg/L sucrose than that of CK and the treatment with 50 g/L sucrose. Crystal violet staining was used to monitor the acclimation of lignin. Different degrees of staining were observed after treatment with 0 sucrose, 20 g/L, and 50 g/L sucrose during AR development. A more intense crystal violet staining was observed after treatment with 50 g/L sucrose than CK and the treatment with 20 g/L sucrose, which indicated that the accumulation of lignin was higher through treatment with 50 g/L sucrose than CK and 20 g/L sucrose treatment. Further, the lowest content of lignin was observed within 6 d of treatment with 20 g/L sucrose (Figure 2).

### 2.3. Determination of Monomer Lignin and Polymer Lignin Contents

The seedlings were treated with 20 mg/L and 50 mg/L sucrose for two days, and the contents of monomer lignin and polymer lignin were determined for the CK0, CK1, GL20, and GL50 libraries (Figure 3a). The seedlings’ monomer lignin was found to be composed of two types of lignin (G and S types), whereas it did not contain the H type. Of the two types of lignin, the content of the G-type lignin was greater than that of the S-type lignin. The lignin content of the G and S types was significantly higher at CK1 than CK0. Seedlings treated with 50 g/L sucrose had a higher lignin content than at CK0, whereas a remarkable decrease was observed after treatment with 20 g/L sucrose (Figure 3b). Further analysis revealed that more than 60% lignin was G-type lignin, and less than 40% lignin was S-type lignin at CK0 and CK1 (20 g/L and 50 g/L sucrose treatment) (Figure 3c). The content of polymer lignin was also determined following the above treatments. The content of polymer lignin was found to increase at CK1 with 50 g/L sucrose and CK0. Further, more content was found at CK1 and 50 g/L sucrose than at CK0. However, the total lignin was significantly decreased after treatment with 20 g/L sucrose (Figure 3d).

### 2.4. The Change in Metabolites Related to Lignin Biosynthesis Owing to Sucrose Treatment

Four libraries, namely CK0, CK1, GL20, and GL50, were constructed to monitor the change in 14 metabolites (p-coumaric acid, p-coumaraldehyde, sinapic acid, sinapyl alcohol, p-coumaryl alcohol, caffeyl alcohol, caffeyl aldehyde, sinapinaldehyde, L-phenylalanine, coniferyl alcohol, caffeate, ferulic acid, 4-hydroxy-3-methoxycinnamaldehyde, and cinnamic acid), which were involved in the synthesis of monomer lignin in lotus seedlings treated with different concentrations of sucrose. Seven metabolites, including sinapinaldehyde and 4-hydroxy-3-methoxycinnamaldehyde, were involved in S-lignin synthesis; coniferyl alcohol, caffeate, and ferulic acid were involved in G-lignin and S-lignin synthesis; L-phenylalanine and cinnamic acid were the primary products for G-lignin and S-lignin synthesis; and six metabolites, such as sinapinaldehyde, 4-hydroxy-3-methoxycinnamaldehyde, coniferyl alcohol, ferulic acid, cinnamic acid, and L-phenylalanine were, were found to increase, and only p-coumaric acid decreased in the CK1/CK0, GL20/CK0, and GL50/CK0 libraries, respectively. Based on the data obtained from these libraries, we further analyzed the altered metabolites in the GL50/GL20 libraries. The content of four important metabolites, namely p-coumaric acid, caffeate, sinapinal aldehyde, and ferulic acid, were found to significantly decrease (Table 1).

### 2.5. Expression of Genes Related to Monomer and Polymer Lignin Contents

All of the genes related to monomer and polymer lignin synthesis were selected from the libraries to further analyze their expression after sucrose and IAA treatment. A total of eight genes involved in monomer lignin synthesis was identified, and the expression of these genes did not change in the CK1/C0, GL20/C0, and GL50/C0 libraries after sucrose treatment (Figure 4a), which indicated that sucrose had no effect on the content of monomer lignin at the transcription level. Genes involved in polymer lignin synthesis were also identified with these concentrations of sucrose treatment. In fact, five genes were found to be expressed at the transcriptional level. The mRNA levels of four genes, including *NnLAC4*, *NnLAC7*, *NnLAC11*, and *NnLAC17*, were increased, and one gene (*NnLAC1*), was decreased after 20 g/L and 50 g/L sucrose treatment (Figure 4b). For these altered genes, including *NnLAC4*, *NnLAC7*, and *NnLAC11*, no remarkable different change in expression was observed between the CK1/CK0 libraries, GL20/C0 libraries, and GL50/C0 libraries. However *NnLAC17* was shown to have a lower transcriptional level in GL20/C0 libraries than that of CK1/CK0 and GL50/C0 libraries (Figure 4b), which indicated that 20 g/L sucrose decreased *NnLAC17* expression. For treatments with 10 μM and 30 μM IAA, *C4H* (*cinnamate 4-hydroxylase*), *CCR* (*cinnayl-CoA reductase*), *CAD* (*cinnamyl alcohol dehydrogenase*), and *F5H* (*ferulate 5-hydroxylase*) related to monomer lignin synthesis altered their expression, whereas *PAL* (*phenylalanine ammonia lysate*) and *COMT* (*cinnamate-O-methyltrasferase*) had no significant changes in the transcriptional level. *CCR* and *CAD* had an increased expression in the A1/C0 and C1/C0 libraries compared to the B1/C0 library, and *F5H* had a greater transcriptional level in the C1/C0 libraries than in the A1/C0 libraries and B1/C0 libraries (Figure 4c). In addition, all of the genes relevant to polymer lignin synthesis were chosen from these libraries. We found that the expression of two genes (*NnLAC4* and *NnLAC17*) involved in polymer lignin synthesis were identified, and *NnLAC17* was found to have an improved transcription level in the C1/C0 libraries compared to the A1/C0 libraries and B1/C0 libraries (Figure 4d).

### 2.6. Cloning and Expression of NnLAC17

*NnLAC17* was cloned from the lotus seedlings and sequenced by Sangon Biotech (Shanghai, China) Co., Ltd. This gene had a full-length sequence of 1749 bp and encoded 583 amino acid residues. The deduced protein of *NnLAC17* showed an 84%, 84%, 83%, and 85% similarity with that of Vitis riparia (XP_034687366.1), Vitis vinifera (XP_002284473.1), Spatholobus suberectus (TKY66042.1), and Trema orientale (PON84167.1), respectively (Additional Figure 1). In addition, a phylogenetic analysis of *LACs* revealed that all *LACs* could be classified into seven groups, and most of the groups included several members, aside from one group, which included only one gene (*NcLAC17*). *SiLAC17*, *CsLAC17*, *NnLAC17*, *PeLAC17*, *PtLAC 17*, *PdLAC17*, and *RcLAC17* were classified into a group. Among those, *CsLAC17* and *NnLAC17* were classified into one sub-group, which indicated that the *LAC17* of lotuses was closely related to that of cucumbers in the gene sequence (Figure 5a).

Different concentrations of IAA and sucrose were applied to lotus seedlings, and the *NnLAC17* expression was analyzed after two days of treatment. No significant difference in *NnLAC17* was observed at the transcription level in control plants and plants treated with 20 g/L sucrose. However, 50 g/L sucrose was found to remarkably enhance *NnLAC17* gene expression (Figure 5b). The same phenomenon was also observed in seedlings treated with IAA. The seedlings treated with 150 µM IAA had higher mRNA levels than the control plants and those treated with 10 µM IAA (Figure 5c). Based on gene expression in different organs, a higher expression of *NnLAC17* was observed in roots than in stems and leaves (Figure 5d).

### 2.7. Functional Analysis of NnLAC17 in Transgenic Arabidopsis Thaliana

*sn1301: NnLAC17* was constructed and overexpressed in *Arabidopsis* plants by the control of a cauliflower mosaic virus 35S promoter to assess gene function. The “positive” plants were identified by hygromycin screening and the RT-PCR technique (Additional Figure 2). The T2 generation of transgenic plants was planted in the base material, and the root length and number were assessed with seedlings at 5–6 leaf age. The plant growth and length of roots in transgenic plants and non-transgenic plants had no significant difference (Figure 6a,b). Further analysis showed that an average of twenty-six roots in each plant was observed in non-transgenic plants at 5–6 leaf age, and approximately twenty roots were found in transgenic plants (Figure 6c), which indicated that the number of roots in transgenic plants decreased compared to non-transgenic plants. In addition, we observed that the content of polymer lignin was markedly increased in transgenic *Arabidopsis* plants after the overexpression of *NnLAC17* (Figure 6d). Compared with non-transgenic plants, *NnLAC17* was found to participate in lignin synthesis, which was involved in the ARs development of lotus seedlings.

## 3. Discussion

In this experiment, we observed that a high concentration (50 g/L) of sucrose inhibited ARs formation, whereas a low concentration (20 g/L) of sucrose promoted ARs development (Figure 1), suggesting that the sucrose signal is involved in ARs formation in the lotus seedlings. The developmental process of ARs in lotus seedlings, including the induction stage, which is the initial formation; the developmental stage of root primordium; and the expressed stage are found [6,30]. These stages are regulated by multiple factors [19,30,31,32]. IAA is involved in the early stages of root formation, and any incidence relevant to IAA synthesis and flux can affect root development [33,34,35]. Cheng et al. (2020) reported that the exogenous application of low concentrations of IAA (10 μM) increases the ARs number, and further analysis reveals an IAA-regulating role at the induction stage and developmental stages in lotus seedlings [32]. Another important regulator of ARs development is sucrose, which has already been found in the plant kingdom. The effect of sucrose occurs throughout the entire stage of ARs development [32]. However, the pathway regulated by sucrose has not been uncovered. Recently, an interaction between sucrose and auxin was found to occur in ARs formation in lotus seedlings [31], suggesting that sucrose and auxin may simultaneously regulate ARs development.

In this study, a microstructural observation of the development of ARs is carried out. Treatment with 50 g/L sucrose results in deeper crystal violet staining in the ARs primordium than that observed at CK and the treatment with 20 g/L sucrose (Figure 2). Such findings suggest that a high correlation may exist between ARs formation and lignin deposition. This phenomenon has also been identified in carrot taproots. Lignin accumulation decreases after the application of exogenous IBA, which results in an increase in root length [36]. Lignin metabolism originates from the phenylpropanoid pathway and participates in various biological processes during plant growth [37,38]. Lignin depletion causes plant dwarfism, which is derived from the disruption of cell wall formation [39]. The silencing of *C4H* can markedly decrease the lignin content and directly result in seedling lethality and an obstacle to leaf development [40]. Lignin deposition is found during root development in sweet potato, suggesting that lignin metabolism is closely correlated with root formation [41]. Further experiment shows that the decreased expression of *CH4* reduces the number of *Arabidopsis* lateral roots relative to AUX transport [42]. Lotus seedlings treated with 50 g/L sucrose were found to have a higher lignin content than at CK. In addition, there was a significant decrease in the lignin content following treatment with 20 g/L sucrose (Figure 3), suggesting that sucrose affects ARs formation derived from the regulation of lignin metabolism. Lignin deposition is related to the age of the plant. Durkovic et al. (2011) reports that 3-year-old plants have higher concentrations of lignin than 1-year-old plants [43]. Further, callus cells in roots have a higher lignin content than meristem [44]. In summary, root development and lignin deposition have a negative correlation, and lignin accumulation negatively affects root development.

Lignin synthesis includes two steps: monomer lignin synthesis and polymer lignin synthesis, and many genes are involved in these processes [45,46]. These biological processes are affected by many factors, including plant hormones, light, stress, temperature, and nitrate [47,48,49,50]. For the synthesis of monomer lignin, *PAL*, *C4H*, *4CL* (*4-coumarate-CoA ligase*), *CCR*, *CAD*, *SAD* (*sinapyl alcoholdehydrogenas*), *COMT,* and *F5H* catalyze the conversion of phenylalanine into p-coumaryl alcohol, sinapyl alcohol, and coniferyl alcohol, resulting in the formation of monomer lignin with the help of POD (*peroxidase*) [23,45,51,52,53,54,55]. The decreased expression of *PALs* and *4CL* reduces lignin accumulation [23,56], and the increased expression of *POD* can significantly improve the lignin content in plants [57]. In this study, we found no difference in the expression of *PAL*, *C4H*, *4CL*, *CCR*, *CAD*, *SAD*, *COMT,* and *F5H* under various concentrations of sucrose (Figure 4). However, the contents of p-coumaric acid, caffeate, sinapinal aldehyde, and ferulic acid were markedly decreased in the GL50/GL20 libraries (Table 1), which indicated that more metabolites were used in the synthesis of polymer lignin after 50 mg/L sucrose treatment. A further analysis was carried out in libraries treated with different concentrations of sucrose and IAA, and all *NnLACs* were selected to monitor the changes in gene expression related to lignin synthesis. We found that only *NnLAC17* showed the same expression tendency; thus, the involvement of *NnLAC17* in polymer lignin synthesis might be the main factor resulting in different lignin contents in lotus seedlings responding to various concentrations of sucrose and IAA.

*LACs* (*Laccases*) play an important role in the polymerization of lignin, which is involved in cell wall formation. In addition, the polymer distribution of lignin is regulated by LACs in gymnosperm compression wood [58]. Recently, *LACs* have been found to be involved in various abiotic stress responses in plants [49]. The overexpression of *LAC2P* can improve water transport, which leads to enhanced plant drought adaptation [59]. H_2_O_2_ (hydrogen peroxide) is reported to be an important signaling molecule. In fact, the exogenous application of H_2_O_2_ promotes AR formation in mung bean seedlings, whereas the expression of *Laccase7*, which is related to lignin synthesis, is found to be significantly reduced [60], suggesting that the formation of ARs and lignin content are opposing processes. Our data further revealed that increased lignin affected the root development in transgenic plants of *Arabidopsis thaliana* after the overexpression of a lotus gene, *NnLAC17* (Figure 6). Such a finding also indicated that *NnLAC17* might be an important gene regulating AR formation in lotuses by improving the lignin content.

Aside from its physiological role in plant growth, lignin also showed great commercial value. In fact, lignin has already been explored and utilized in the food industry, and is now considered as a valuable and renewable material for future food produce [61,62].

Zhang et al. (2013) have reported that lignin not only acts as a dietary fiber, but that it can also promote the activity of α-amylase, which directly leads to a decrease in the α-helical content, and improves the content of the polarity around tryptophan residues and protein granules [63]. A similar pheromone was also found, where lignin can also interact with wheat gluten to make protein soluble through the activity of aromatic hydroxyl antiradicals [64]. In lotuses, some kinds of organs, such as the lotus seed, leaf, and belt, have already been utilized to produce health foods, drinks, and even medicine (some value is derived from lignin) [32]. In this study, we found that the overexpression of *NnLAC17* can improve the content of the transgenic Arabidopsis plant (Figure 6). Therefore, improving the lignin content by genetic engineering means that it is a probable available pathway to make food more nutritious and healthier in lotuses in the future.

## 4. Materials and Methods

### 4.1. Plant Growth

The lotus seed species, Taikong36, was selected for all experiments in this study. The seeds used in this study were derived from the open field of Yangzhou University. In spring (usually in April), the lotus was planted in the field for germination with moist soil. After the petiole broke through the soil surface, the water depth was maintained at 5–10 cm; the deeper water (20–40 cm) was required for the development of the plant and the available temperature was maintained between 20 and 30 °C during the entire growth season. The application of fertilizer and pest control was carried out in the same manner employed in the conventional management of the field. The seeds were harvested in November and placed in a container at normal temperature.

### 4.2. The Role of Sucrose in ARs Formation

The seed coat of lotus was punched to enable the uptake of available water (for approximately three days) and placed at 26 °C for approximately two days under dark conditions. Fifty germinated seedlings from each treatment were selected and placed into 0 g/L, 20 g/L, and 50 g/L sucrose solutions for two days. The germination rates and numbers were counted on day 4 after transfer into water for 2 d. At the same time, the shoot tips (0.5 cm) of lotus were sterilized with 75% ethanol for 20 s, and then transferred into sodium hypochlorite (10%) for 20 min. All sterilized shoot tips were firstly cultured on the MS culture medium (MS + 30 g sucrose + 0.1% NaOH + 7 g/L agar + 0.5 mg/L KT (kinetin) + 0.1 mg/L NAA (naphthalic acetic acid)) for approximately 30 days, and then were transferred to medium containing (MS + 0%, 20%, or 50% sucrose + 0.1% NaOH + 7 g/L agar + 0.5 mg/L KT + 0.1 mg/L NAA) to continue cultivation for approximately 40 day, and the adventitious roots were counted. The temperature was 26 °C with 12 h light and 12 h dark–light cycle. All statistical data were expressed as the mean ± SE of three repetitions of experiments. The SPSS software ver. 14.0 (SPSS Inc., Chicago, IL, USA) was used for statistical analyses.

### 4.3. Observation of the Paraffin Sections

Seed treatment before germination and germination condition was the same as above. The treated seedlings with 20 and 50 mg/L sucrose were selected, and the hypocotyls of seedlings at 0, 2, 4, and 6 days after treatment were cut into small pieces of 2.5 mm × 2.5 mm × 2 mm (length, width, and height), and then placed in a container filled with FAA fixing fluid (solution amount was 20-fold that of samples).

The container with fixed samples was produced in a vacuum state using a syringe for 5 s. The lid of the container was opened for gas exchange after five min.

This process was repeated in triplicate. The container was transferred to a clean bench at normal temperature for 24 h. Thereafter, 50%, 70%, 85%, 95%, and 100% ethanol were applied to dehydrate the samples for approximately 25–30 min. After dehydration with ethanol, the samples were treated with a mixed solution (one-half volume of ethanol and one-half volume of xylene and pure xylene) for approximately 25–30 min. Paraffin debris was prepared and thawed on an electric stove and then poured into the container with the samples for 24 h. The tissues embedded in paraffin were cut into small blocks, and 10 µm of wax tape was prepared using a slicer. The wax tapes were placed on a glass slide, and then, respectively, transferred into pure xylene, mixed solution (one-half volume of pure xylene and one-half volume of absolute ethanol), and absolute ethanol for 10 min. The slide was dried at room temperature, and 2–3 drops of 0.75% crystal violet solution were placed on the samples for 5 min. The samples were washed with distilled water and then observed using an optical microscope.

### 4.4. Determination of Monomer Lignin and Polymer Lignin Contents

The hypocotyls of lotus seedlings treated with 20 mg/L and 50 mg/L for two days were used as materials to determine the monomer lignin and polymer lignin contents (CK0: 0 day of treatment; CK1: three days without treatment; GL50: three days with 50 g/L sucrose treatment; GL20: three days with 20 g/L sucrose treatment). For monomer lignin analysis, ten milligrams of dried sample were added to a mixture (2.5% boron trifluoride, 10% ethanethiol, 87.5% dioxane) for 4 h in a metal bath followed by 300 µL of 0.4 m sodium bicarbonate. The mixture was vibrated, and 2 mL water and 0.3 mL ethyl acetate were added. The mixture was centrifuged at 14,000 rpm for 10 min at normal temperature, and the supernatant was retrieved. The supernatant was dried with nitrogen, and 150 µL pyridine (including an internal standard) and 50 µL trimethylsilane were allowed to react at 60 °C for 1 h. The mixture was centrifuged at 10,000 rpm for 10 min at normal temperature, and the supernatant was retrieved for lignin identification. The lignin content was determined using a GC-MS instrument (7820A-5977B) from Agilent Technologies Inc. (Palo Alto, CA, USA). The conditions for chromatography (mass spectrometry) and quantitative analysis lignin were based on a previous protocol [65].

For polymer lignin analysis, 50 mg of lotus seedlings dry powder was added into 1 mL of 70% ethanol with thorough vortex, and then centrifuged. The supernatant was discarded, and 1 mL of chloroform/methanol (1:1 *v*/*v*) solution was added. After centrifuge, the supernatant was also discarded. A total of 1 mL acetone was added to wash precipitant, and then vacuumed for drying. A total of 1.5 mL sodium acetate buffer (0.1 M, Ph5.0) was put into tube with precipitant, and then heated at 80 °C for 20 min. Ten microliters of sodium azide (0.01%) amylase and pullulanase were added and incubated overnight at 37 °C. The reaction was terminated by heating at 100 °C condition, and the supernatant was discarded by centrifugation at room temperature. Precipitation was cleaned by adding H_2_O, and then 1 mL acetone was added. The mixture was vacuumed for dry acetone, and 1 mg precipitate was chosen to be dissolved by 100 µL acetyl bromide (25%) solution under 50 °C for 3 h. Then, 400 µL sodium hydroxide (2 M) and 70 µL hydroxylamine hydrochloride (0.5 M) were added and thoroughly mixed. The volume of mixture was fixed to 2 mL with acetic acid, and 200 µL supernatant was used to measure absorbance value at 280 nm by multifunctional microplate reader (1510-04201, Biotek, Winooski, VT, USA). Polymer lignin content was counted using following formula:$$\mathrm{Lignin}\;\mathrm{content}\;=\frac{\mathrm{ABS}}{\mathrm{Coeff}\;\times \;0.539\;\mathrm{cm}}\times \frac{2\;\mathrm{ml}}{\mathrm{Weight}}\times 100\%$$

ABS: light absorption value; Coeff: absorption coefficient.

In addition, the roots of six-leaf stage seedlings of transgenic Arabidopsis plants and none of the transgenic plants were collected for polymer lignin identification. The material treatment and identification method was the same as that mentioned above.

### 4.5. Metabolite Analysis of Lignin during ARs Formation

The hypocotyls of control lotus seedlings and seedlings treated with 20 g/L and 50 g/L for two days were selected for metabolites analysis. All freeze-dried hypocotyls were crushed using a mixer mill (MM 400, Retsch, Shanghai, China) with a zirconia bead for 1.5 min at 30 Hz. Powder (100 mg) was weighed and extracted for approximately 12 h at 4 °C with 1.2 mL 70% aqueous methanol. Following centrifugation at 10,000× *g* for 10 min at normal temperature, the extracts were absorbed (CNWBOND Carbon-GCB SPE Cartridge, 250 mg, 3 mL; ANPEL, Shanghai, China, www.anpel.com.cn/cnw (accessed on 22 April 2021)) and filtered (SCAA-104, 0.22 μm pore size; ANPEL, Shanghai, China, http://www.anpel.com.cn/ (accessed on: 16 May 2021)) before UPLC-MS/MS analysis.

### 4.6. UPLC (Ultra Performance Liquid Chromatography) Conditions

The sample extracts were analyzed using a UPLC-ESI-MS/MS system (UPLC, Shim-pack UFLC SHIMADZUCBM 30A system, www.shimadzu.com.cn/ (accessed on: 23 May 2020); MS, Applied Biosystems 6500 QTRAP, www.appliedbiosystems.com.cn (accessed on 23 May 2020). The analytical conditions were as follows: column, Waters ACQUITYUPLCHSST3C18 (1.8 µm, 2.1 mm × 100 mm); and mobile phase, solvent A- pure water with 0.04% acetic acid and solvent B- acetonitrile with 0.04% acetic acid. Sample measurements were performed with an Agilent program that employed the starting conditions of 95% A and 5% B. Within 10 min, a linear gradient to 5% A and 95% B was employed, and a composition of 5% A and 95% B was maintained for 1 min. Subsequently, a composition of 95% A and 5.0% B was employed for 0.10 min and retained for 2.9 min. The column oven temperature was set to 40 °C, and the injection volume was 2 μL. The effluent was alternatively connected to an ESI-triple quadrupole-linear ion trap (QTRAP)-MS.

### 4.7. ESI-QTRAP-MS/MS

LIT and triple quadrupole (QQQ) scans were acquired on a triple quadrupole-linear ion trap mass spectrometer (Q TRAP), API 6500 Q TRAP UPLC/MS/MS system, equipped with an ESI Turbo Ion Spray interface, operating in positive and negative ion mode and controlled by Analyst 1.6.3 software (AB Sciex). The ESI source parameters were as follows: ion source, turbo spray; source temperature, 550 °C; ion spray voltage (IS), 5500 V (positive ion mode)/−4500 V (negative ion mode); ion source gas I (GSI), gas II (GSII), and curtain gas (CUR) 50, 60, and 30.0 psi, respectively; and collision gas (CAD), high. Instrument tuning and mass calibration were performed with 10 and 100 μmol/L polypropylene glycol solutions in the QQQ and LIT modes, respectively. QQQ scans were acquired as MRM experiments with a collision gas (nitrogen) set at 5 psi. DP and CE for individual MRM transitions were performed with further optimization of DP and CE. A specific set of MRM transitions was monitored for each period, according to the metabolites eluted within this period.

### 4.8. RNA-seq Analysis of the Differentially Expressed Genes

For the analysis of differentially expressed genes, all genes related to monomer and polymer lignin synthesis were selected and their expression profiles were monitored after treatment with IAA (10 μM and 150 μM) and sucrose (20 g/L and 50 g/L) during AR formation. In the previous study, four libraries treated with 10 μM and 150 μM of IAA (CK library: germinating seeds without treatment; A library, initial AR stages: 2–3 days of seedlings with 10 μM IAA treatment; B library: 2–3 days of seedlings without IAA treatment; C library: 2–3 days of seedlings with 150 μM IAA treatment) and sucrose (C0 library: the samples were collected at day 0; CK1, GL20, and GL50 libraries: the samples were collected at day 1 after treatment with 0, 20, and 50 g/L sucrose respectively) were constructed, and the data have been published by Cheng et al. (2018a) and Cheng et al. (2020a), respectively [19,32]. These data include the samples of the CK library (germinating seeds without treatment), A library (initial AR stages: 2–3 days of seedlings with 10 μM IAA treatment), B library (2–3 days of seedlings without IAA treatment), and C library (2–3 days of seedlings with 150 μM IAA treatment). Genes related to the synthesis of lignin, including monomer and polymer lignin, were selected from above libraries for further analysis.

### 4.9. Cloning and Sequence Analysis of NnLAC17

The sequence of *NnLAC17* was derived from the NCBI database according to the gene expression profile under sucrose treatment. Lotus RNA was extracted from the hypocotyls of three old seedlings using a plant RNA (Ribonucleic Acid) extract mix (Tiangen, Beijing, China). DNase was applied to eliminate residual DNA (Deoxyribonucleicacid) before cDNA synthesis, which was performed according to the protocol of the mix kit (Promega, Madison, WI, USA). A total of 20 μL of PCR (Polymerase Chain Reaction) reaction mixture, which included 2.5 μL dNTP, 2 μL forward and reverse primers, 2.5 μL MgCl_2_, 0.5 μL Taq polymerase, 2 μL cDNA fragments, and 9.5 μL dH_2_O, was used. The PCR program included 35 cycles: 94 °C for 1 min, 94 °C for 1 min, 60 °C for 1 min, 72 °C for 1 min, and 72 °C for 10 min. Forward primer: 5-ATGGGTTCCTTTGTTCTTCC-3; reverse primer: 5-AGGCGGTAGTTTCTGATTTG-3. The plasma of DNA plasmid was sent to Sangon Biotechnology Co., Ltd. (Shanghai, China) for sequencing. For sequence analysis, five LACs, including lotus, vitis riparia, vitis vinifera, spatholobus suberectus, and trema orientale, was used for alignment analysis by DNAman software. In addition, thirty-four *LACs* from different plants were selected, and MEGA-X software was used for phylogenetic tree construction.

### 4.10. Expression Analysis of NnLAC17 in Lotus

The expression profile of *NnLAC17* was determined using qRT-PCR (Quantitative Reverse Transcription PCR). The hypocotyls of seedlings treated with sucrose (0, 20, and 50 g/L) and IAA (0, 10, and 150 μM) were selected at 4 days after being transferred into water for transcriptional level analysis. For the expression of *NnLAC17* in different organs, the leaf, stem, and hypocotyls of seedlings at 6 days (cultivation in water) were selected to monitor expression changes. Total plant RNA was derived from these samples, and DNase was used to eliminate residual DNA. The primer was designed based on the *NnLAC17* sequence using the primer 5.0 software. Forward primer was 5-GGGTTCCTTTGTTCTTCCA-3, and reverse primer was 5-GCCAGTGCAAGGTGATATT-3. *NnActin* was used as the internal standard. The *NnActin* forward primer was 5′-ACGCGTATGAAGTCAGTTGT-3′ and reverse primer was 5′-TTTATGGGGATCAGCTGGT-3′. A 25 μL reaction mixture was prepared, which contained 12.5 μL SYBR Premix Ex Taq II (Tli RNaseH Plus) (2×), 1 μL of each of the forward and reverse primers, 2 μL cDNA, and 8.5 μL dH_2_O. The PCR program consisted of 30 s at 94 °C, followed by 40 cycles of 95 °C for 5 s and 60 s at 60°. Three biological replicates were carried out in this experiment. For data analysis, the 2^−∆∆Ct^ method was used to identify *NnLAC17* expression. The ∆Ct value was obtained according to the Ct (target) and Ct (actin) values in treated plants (∆Ct (target)) and control (∆Ct (normal), and the ∆∆Ct value was calculated according to the data of ∆Ct (target) and ∆Ct (normal). Based on the ∆∆Ct value, the 2^−∆∆Ct^ value was determined.

### 4.11. Vector Construction

*NnLAC17* was cloned and inserted into a clone vector (pGEM-T). pGEM-T was transformed into *Escherichia coli* to expand reproduction. After identification, positive clones were cultured in LB culture medium. The plasmid was extracted from bacterial fluid and digested with *BamHI* and *KpnI* enzymes. Thereafter, the digestion product (sequence of *NnLAC17*) was inserted into the plant transformation vector with a CaMV 35S promoter. The pSN1301:: *NnLAC17* plasmid was inserted into the *Agrobacterium tumefaciens* strain, GV3101. The floral dip method [66] was employed to transform *NnLAC17* into the wild-type Arabidopsis plant. The transformed Arabidopsis was cultured in the glasshouse, and seeds were harvested after approximately three months of growth. Sterilized seeds of the T0 generation were tiled on MS medium containing 20 μg·g^−1^ hygromycin B to select ‘positive’ plants. The temperature of the chamber was 22 °C with 12 h light and 12 h dark–light cycle. In addition, RT-PCR (Reverse Transcription PCR) method was applied for further identification, and the primers were: forward primer was 5-GGGTTCCTTTGTTCTTCCA-3, and reverse primer was 5-GCCAGTGCAAGGTGATATT-3. Reaction mixture and program process was the same as gene expression. Base on the information of data, three lines of positive plants were selected for further study.

### 4.12. Identification of NnLAC17 Function in Arabidopsis Plants

The seeds of transgene (T2 generation) and wild-type plants were cultured in pots, and seven-leaf-old transgenic seedlings and wild-type plants were selected for functional analysis. Further, the seeds of the transgene (T2 generation) and wild-type seeds were sterilized with 75% alcohol for 20 s and transformed into sodium hypochlorite (30%) for 20 min. The sterilized seeds were sown on the medium, and root number and length was counted at six-leaf age of seedlings. For statistical analysis, all data are presented as the mean ± SE of triplicate independent treatments, with approximately 10 seedlings per experiment, and approximately thirty seedlings being used for three biological replicates. The SPSS software ver. 14.0 (SPSS Inc., Chicago, IL, USA) was used for data analysis.

### 4.13. Statistical Analysis

Statistical analyses were performed using the SPSS software ver. 14.0 (SPSS Inc., Chicago, IL, USA). Approximately twenty plants in each experiment were used for statistical analysis. The data were recorded as the means ± SE of three experiments, and the differences at *p* < 0.05 were accepted as the level of significance.

## 5. Conclusions

In this study, we found that lignin synthesis was strictly regulated by sucrose according to the change in monomer and polymer lignin content. In addition, the microstructural observation and intermediate metabolites of lignin confirmed that lignin synthesis is involved in the sucrose signal transduction pathway in the seedlings of lotuses. Based on the above result, genes involved in lignin synthesis (monomer and polymer lignin) were identified in the libraries treated with different concentrations of sucrose. *NnLAC17* was selected and considered as an important gene in lignin synthesis, according to gene expression, in responding to sucrose and IAA treatment. Further experiments, including spatiotemporal expression and transgenic *Arabidopsis thaliana*, uncovered that lignin synthesis, which is affected by *NnLAC17*, is a key factor involved in the ARs formation of lotus seeding.

## Figures and Tables

**Figure 1 ijms-23-02250-f001:**
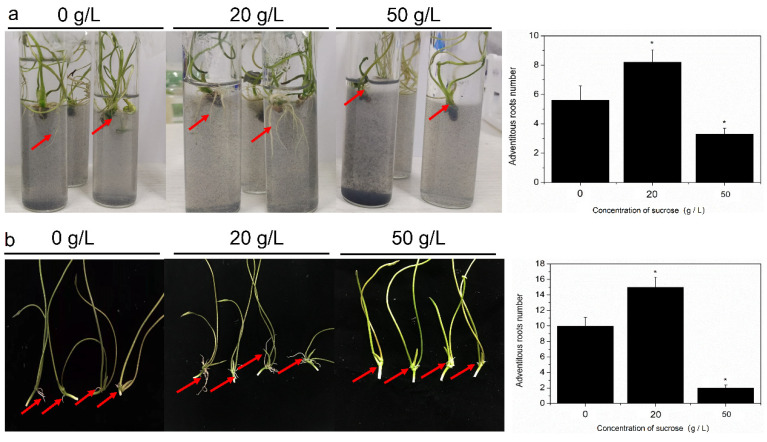
The influence of 20 g/L and 50 g/L sucrose on ARs formation in lotus seedlings. (**a**) AR formation in the water; (**b**) AR development in the MS medium (the data of (**b**) were referred by Cheng et al. 2020 [19]). (Red arrow represents point of the growth of adventitious root; “*” represents significant difference, *p* < 0.05.)

**Figure 2 ijms-23-02250-f002:**
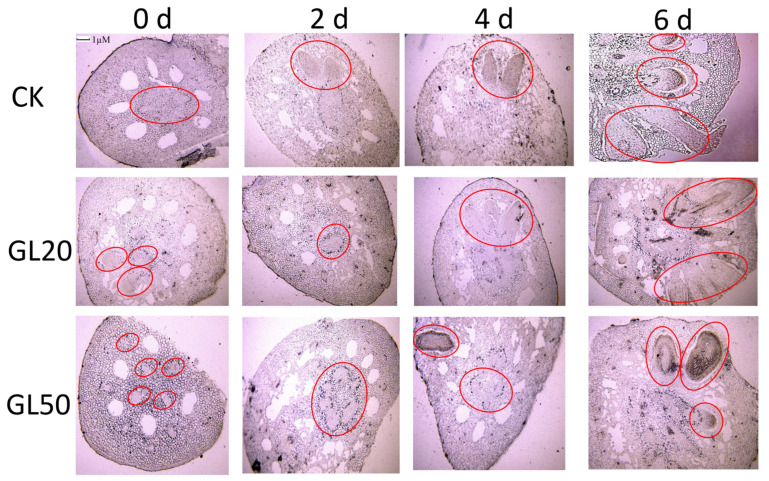
The change in lignin synthesis based on an observation of the microstructure during AR formation in lotus seedlings treated with 20 g/L and 50 g/L sucrose. GL20 represents 20 g/L treatment and GL50 represents 50 g/L treatment. The seedlings were firstly treated with 20 g/L and 50 g/L sucrose, and then transferred into water for continue growth. The hypocotyls at 0 days, 2 days, 4 days, and 6 days after transfer were collected for microstructure observation.

**Figure 3 ijms-23-02250-f003:**
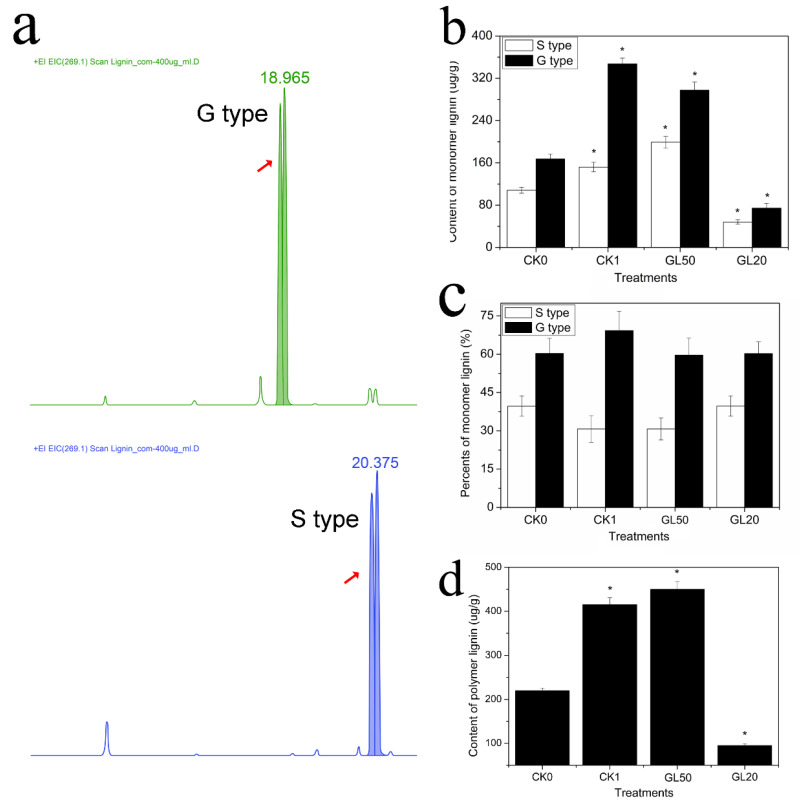
Identification of the lignin content in lotus seedlings treated with 20 g/L and 50 g/L sucrose during ARs formation. (**a**) The lignin was analyzed by ESI-QTRAP-MS/MS (Electrospray Ionization Quadrupole Ion TRAP Mass Spectrum/Mass Spectrum). (**b**) Determination of the monomer lignin content in the CK0, CK1, GL50, and GL20 libraries. (**c**) Determination of the monomer lignin types in the CK0, CK1, GL50, and GL20 libraries. (**d**) Analysis of polymer lignin content in the CK0, CK1, GL50, and GL20 libraries. (Red arrow represents acquisition time of G-type and S-type lignin, “*” represents significant difference at 0.05 level, *p* < 0.05.)

**Figure 4 ijms-23-02250-f004:**
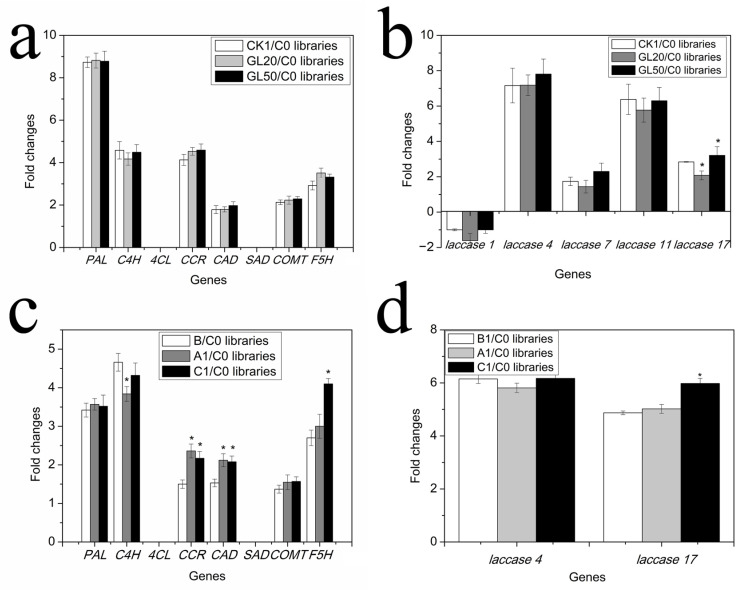
The expression of genes related to lignin synthesis during ARs formation in lotus seedlings. (**a**) The expression of genes related to monomer lignin synthesis in the GL20 and GL50 libraries. (**b**) The expression of genes related to polymer lignin synthesis in the C0, CK1, GL20, and GL50 libraries. (**c**) The expression of genes related to monomer lignin synthesis in the C0, A1, B, and C1 libraries. (**d**) The expression of genes related to polymer lignin synthesis in the C0, A1, B, and C1 libraries. (“*” represents significant difference at 0.05 level, *p* < 0.05.)

**Figure 5 ijms-23-02250-f005:**
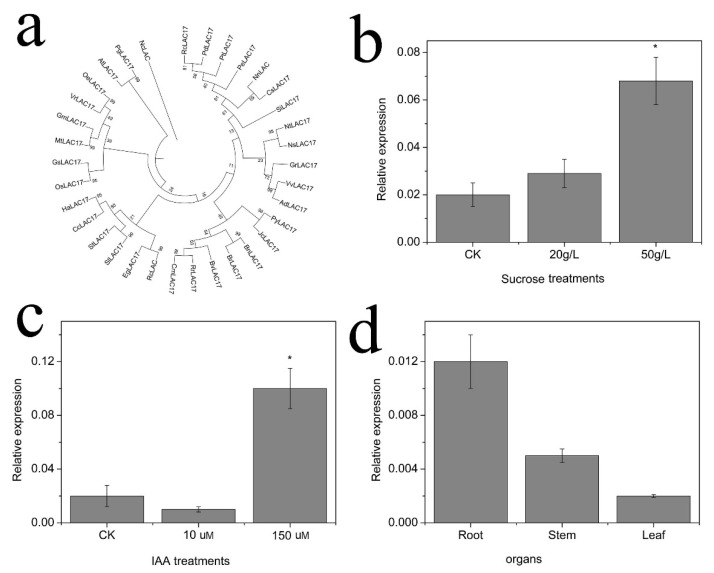
Cloning and expression analysis of *NnLAC17* in lotus seedlings. (**a**) Analysis of the similarity between *NnLAC17* and that of other species. (**b**) Relative expression of *NnLAC17* in seedlings treated with 20 g/L and 50 g/L sucrose. (**c**) Relative expression of *NnLAC17* in seedlings treated with 10 µM and 150 µM IAA. (**d**) Relative expression of *NnLAC17* in different organs of lotuses. (“*” represents significant different at 0.05 level, *p* < 0.05.)

**Figure 6 ijms-23-02250-f006:**
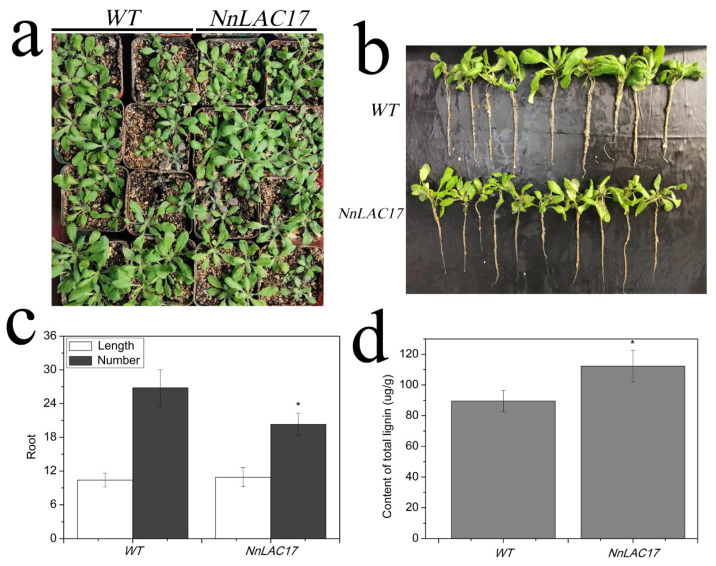
Functional analysis of *NnLAC17* in transgenic *Arabidopsis thaliana*. (**a**) Phenotypic investigation of *NnLAC17* transgenic plants and wild-type plants at 5–6 leaf age. (**b**) Comparative analysis of root length and number of transgenic plants and wild-type plants. (**c**) Determination of root length and number in *NnLAC17* transgenic plants and wild-type plants (*WT*). (**d**) Investigation of polymer lignin in *NnLAC17* transgenic plants and wild-type plants. (“*” represents significant difference at 0.05 level, *p* < 0.05; *WT* represents non-transgenic plants, and *NnLAC17* represents transgenic plants with *NnLAC17*).

**Table 1 ijms-23-02250-t001:** Analysis of the metabolism of monomer lignin in lotus seedlings treated with 20 g/L and 50 g/L sucrose (NA means that the metabolites are too little to be determined; “--------” represents no difference between two libraries).

Compounds	*p* Value	Fold Change	Types
CK1/CK0 libraries			
p-Coumaric acid	0.0112	0.2743	Down-regulated
p-Coumaraldehyde	NA	NA	
Sinapic acid	NA	NA	
Sinapyl alcohol	NA	NA	
p-Coumaryl alcohol	NA	NA	
Caffeyl alcohol	NA	NA	
Caffeyl aldehyde	NA	NA	
Sinapinaldehyde	0.0015	3.4747	Up-regulated
L-Phenylalanine	0.0052	1.8344	Up-regulated
Coniferyl alcohol	0.0225	23.0678	Up-regulated
Caffeate	0.0305	1.4309	Up-regulated
Ferulic acid	0.0564	1.6682	Up-regulated
4-Hydroxy-3-methoxycinnamaldehyde	0.0019	10.5479	Up-regulated
Cinnamic acid	0.0036	2.5194	Up-regulated
GL20/CK0			
p-Coumaric acid	0.0407	0.1107	Down-regulated
p-Coumaraldehyde	NA	NA	
Sinapic acid	NA	NA	
Caffeate	0.0369	1.156	
Sinapyl alcohol	NA	NA	
p-Coumaryl alcohol	NA	NA	
Caffeyl alcohol	NA	NA	
Caffeyl aldehyde	NA	NA	
Sinapinal dehyde	0.0147	7.2121	Up-regulated
L-Phenylalanine	0.0497	1.6486	Up-regulated
Coniferyl alcohol	0.01762	19.5263	Up-regulated
Ferulic acid	0.0113	2.3898	Up-regulated
4-Hydroxy-3-methoxycinnamaldehyde	0.0415	16.7289	Up-regulated
Cinnamic acid	0.02470	2.1252	Up-regulated
GL50/CK0			
p-Coumaric acid	0.04766	0.079	Down-regulated
p-Coumaraldehyde	NA	NA	
Sinapic acid	NA	NA	
Caffeate	0.0325	0.899	
Sinapyl alcohol	NA	NA	
p-Coumaryl alcohol	NA	NA	
Caffeyl alcohol	NA	NA	
Caffeyl aldehyde	NA	NA	
Sinapinal dehyde	0.0057	5.2929	Up-regulated
L-Phenylalanine	0.0073	1.5573	Up-regulated
Coniferyl alcohol	0.0072	20.4520	Up-regulated
Ferulic acid	0.03586	1.9201	Up-regulated
4-Hydroxy-3-methoxycinnamaldehyde	0.0001	17.3515	Up-regulated
Cinnamic acid	0.0004	2.4017	Up-regulated
GL50/GL20			
p-Coumaric acid	0.0386	0.7140	Down-regulated
p-Coumaraldehyde	NA	NA	
Sinapic acid	NA	NA	
Caffeate	0.0339	0.7823	Down-regulated
Sinapyl alcohol	NA	NA	
p-Coumaryl alcohol	NA	NA	
Caffeyl alcohol	NA	NA	
Caffeyl aldehyde	NA	NA	
Sinapinal dehyde	0.0157	0.7338	Down-regulated
L-Phenylalanine	0.0014	--------	
Coniferyl alcohol	0.0082	--------	
Ferulic acid	0.0254	0.7034	Down-regulated
4-Hydroxy-3-methoxycinnamaldehyde	0.0497	--------	
Cinnamic acid	0.0145	--------	

## Data Availability

The materials of all of the experiments were supported by aquatic vegetable Labof Yangzhou University. The collection of seeds complied with local and national guidelines and permissions of seeds were obtained. The detailed data derived from sucrose treatment have been deposited in the NCBI database (Bioproject number). All of the raw tags derived from IAA treatment were deposited in the National Center for Biotechnology Information (BioProject ID: PRJNA398315; accession number: SRR5944803–SRR5944814).

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
