# Peer review of "Lignin Synthesis, Affected by Sucrose in Lotus (Nelumbo nucifera) Seedlings, Was Involved in Regulation of Root Formation in the Arabidopsis thanliana"

_ijms, 2022, doi:10.3390/ijms23042250_

Round 1
Reviewer 1 Report
The manuscript is certainly of interest to readers, however, to improve its quality, I propose to make a number of corrections.
- The text of the manuscript contains a lot of repetitions. For example, in the Summary section, the text on lines 15-16 is identical in meaning to the text on lines 31-32. In the Introduction section, the third sentence can be deleted because it in its meaning repeats the text of the first and second sentences. The beginnings of the three sections on page 12 ("The role of sucrose in ARs formation", "Observation of the paraffin sections", and "Determination of monomer lignin and polymer lignin contents") contain the same information. I strongly recommend that the text of the ENTIRE manuscript should be revised.
- It is necessary to decipher the abbreviations NAA, KT, IBA.
- Line 342. It is not clear how many grams of sucrose were added to MS medium.
- Line 343. I recommend writing "30 days", not "30 d".
- Line 397. “…extracted overnight at 4°C…”. I recommend that you clearly indicate the number of hours.
- There are many comments on the drawings:
- in the captions for ALL figures there is no decoding of symbols. For example, what do the red arrows in fig.1 point to? What do the two little asterisks above the standard errors mean? What is 0d, 2d, 4d, 6d in fig.2, etc.?
- in fig. 1 it is not indicated where is the figure “a”, and where is the figure “b”;
- in fig. 1 on the graphs, it is better to sign the abscissa axis “Concentration of sucrose, g / L” and remove the symbols next to the numbers;
- what do the areas highlighted in red in fig.2 represent?
- why did the authors in the caption to fig.3 write “Analysis of polymer lignin content…”, and in fig.3d write "Content of total lignin"?
- in fig.3b, c, d, the treatments are designated as CK0 and CK1, and in the caption to fig.3 they are marked as C0 and C1;
- in fig.4a and b are labeled “CK1/C0 libraries”, but there is no mentions of CK1 in the caption to fig.4;
- in fig.5c, the abscissa is labeled “IAA treatments” and the data are labeled in umol/L, but the figure caption reads “Relative expression of NnLAC17 in seedlings treated with 10 g/L and 150 g/L sucrose”.
- What is "P lignin" in “The change in metabolites related to lignin biosynthesis owing to sucrose treatment” section?
- What is GL60 and NA in table 1?
- Line 239-240. The names of the figures are mixed up.
- Information about statistical analysis is not presented as a separate subsection and is described лучше too briefly.
Author Response
Dear editor and reviewers:
All the authors showed many thanks to the editor and reviewers and suggestion to handle this text file. Every one of co-authors was degree with comments derived from reviewers, and the manuscript was revised according to the suggestions:
Reviewer 1
Question 1
The text of the manuscript contains a lot of repetitions. For example, in the Summary section, the text on lines 15-16 is identical in meaning to the text on lines 31-32. In the Introduction section, the third sentence can be deleted because it in its meaning repeats the text of the first and second sentences. The beginnings of the three sections on page 12 ("The role of sucrose in ARs formation", "Observation of the paraffin sections", and "Determination of monomer lignin and polymer lignin contents") contain the same information. I strongly recommend that the text of the ENTIRE manuscript should be revised.
Response 1
The entire manuscript was carefully revised for grammar and sentences. At the same time the repetitive sentences were also deleted according to the reviewer’s suggestion. The tracked version of text file was provide.
Question 2
It is necessary to decipher the abbreviations NAA, KT, IBA.
Response 2
I have added the detail illustration for these hormone according to reviewer’s suggestion in the text file.
Question 3
Line 342. It is not clear how many grams of sucrose were added to MS medium.
Response 3
I am sorry not to clearly explain in this section, and I have revised it in the manuscript. Actually, the entire process was classified into two steps. In the first step, the sterilized shoot tips were cultured on the MS culture medium (MS+ 30 g sucrose + 0.1% NaOH + 7g/L agar + 0.5 mg/L KT (kinetin) + 0.1 mg/L NAA (naphtalic acetic acid)) for about 30 d; In the second step, the tips were transferred to the MS medium containing 0, 20, and 50 g/L sucrose, and other components were the same as above mentioned.
(MS+ 0 %, 20% or 50% sucrose + 0.1% NaOH + 7 g/L agar + 0.5 mg/L KT + 0.1 mg/L NAA).
Question 4
Line 343. I recommend writing "30 days", not "30 d".
Response 4
I have revised style of writing, and the entire manuscript was carefully checked and revised according to the reviewer’s suggestion.
Question 5
Line 397. “…extracted overnight at 4°C…”. I recommend that you clearly indicate the number of hour
Response 5
I had revised this mistake, and this process will use up about 8-12 hours. In this study, about 12 hours was used, so in the text file, I have replace “overnight” with “about 12 hours”.
Question 6
There are many comments on the drawings:
Response 6
I have revised the manuscript according to the reviewer’s commands, and the detail revise was shown in “tracked copy” of manuscript.

Reviewer 2 Report
This study revealed the involvement of sucrose metabolism in AR development and lignin synthesis in the regulation of ARs formation. The authors further monitored a gene (NnLAC17) related to lignin synthesis and found that the expression of the gene was altered after plants were treated with 20 and 50 g/L sucrose. The paper presented interesting results with a novel methodology. However, the use of 20 to 50 g/L sucrose seems to be not economical considering the yield of obtained lignin monomer and polymer in this study. The novelty of the paper must also be highlighted concerning already existing material by the same authors (DOI: 10.21203/rs.3.rs-849586/v1). Considering together, the major revisions are recommended based on the following comments:
- What is the yield of lignin synthesis in this study at the presence of 50 g/L sucrose? The authors must add one new section and discuss this process from the economic point of view.
- The abstract and conclusions seem similar. One of them should be modified. Please follow the journal guideline with a maximum of 200 words.
- Keywords: Please replace LC-MS/MS with other relevant ones.
- Please highlight the novelty and objective of the paper in the last paragraph of the Introduction.
- Line 81-89: This paragraph must be completely changed as it reflects the conclusion rather than the objectives and novelty statement.
- A similar paper with the same content is already on the internet published: https://www.researchsquare.com/article/rs-849586/v1. Please make sure that this paper is not the “salami-slicing” of the previous paper.
- Line 387: it is necessary to give details about the quantitative analysis of lignin.
- Line 388: “Three biological replicates were carried out in this experiment” this sentence is irrelevant to this section. It must be removed.
- Could the authors make sure that the English of the written title sounds perfectly correct?
Author Response
Dear editor and reviewers:
With many thanks, authors really appreciate the editor and reviewers’ helpful comments and valuable suggestions. According to the comments, we have made following corrections and modifications for the text file:
Question 1
What is the yield of lignin synthesis in this study at the presence of 50 g/L sucrose? The authors must add one new section and discuss this process from the economic point of view.
Response 1
The content of polymer lignin in the hypocotyl of lotus seedlings was 450.2 µg/g (per 1 g dry samples of hypocotyl contains 450.2 µg polymer lignin). The lignin is very important for lotus due to its function to maintain vertical growth of petiole, sustain large leaf for petiole. Until now, commercial exploitation has carried out to explored lotus value from economic point (some value was from lignin). In the discussion section of manuscript, we added the related content.
Question 2
The abstract and conclusions seem similar. One of them should be modified. Please follow the journal guideline with a maximum of 200 words.
Response 2
Thanks a lot for reviewer’s suggestion, and I have revised the conclusion section. in addition, the abstract section was also revised, and made the section less than 200 words
Question 3
Keywords: Please replace LC-MS/MS with other relevant ones
Response 3
I have replaced the word of LC-MS/MS with sucrose due to its role in regulating lignin synthesis and ARs development.
Question 4
Please highlight the novelty and objective of the paper in the last paragraph of the Introduction.
Response 4
I have carefully revised the last paragraph of introduction section according to reviewer’s suggestion.
Question 5
Line 81-89: This paragraph must be completely changed as it reflects the conclusion rather than the objectives and novelty statement.
Response 5
I have completely changed the last paragraph (line 81-89) with suggestion of reviewer.
Question 6
A similar paper with the same content is already on the internet published: https://www.researchsquare.com/article/rs-849586/v1. Please make sure that this paper is not the “salami-slicing” of the previous paper.
Response 6
Thanks a lot for the reviewer’s alarming, and I have carefully checked the version of manuscript existing on internet. This version online was almost the same with this submission, and while it was not the “salami-slicing” of the previous paper. The text file submitted to international journal of molecular science was previously submit to other journal, and the copy (https://www.researchsquare.com/article/rs-849586/v1) was only used for review, and this process was completed. Therefore, I am sure that the same paper, even relevant content has not published in other journal, aside from international journal of molecular science.
Question 7
Line 387: it is necessary to give details about the quantitative analysis of lignin.
Response 7
I have added the details methods and quantitative analysis of lignin
Question 8
Line 388: “Three biological replicates were carried out in this experiment” this sentence is irrelevant to this section. It must be removed.
Response 8
I have deleted this sentence according to reviewer’s suggestion.
Question 9
Could the authors make sure that the English of the written title sounds perfectly correct?
Response 9
Thanks! The title, which was revised into “lignin synthesis, affected by sucrose in lotus (Nelumbo nucifera) seedlings was involved in regulation of root formation in the Arabidopsis thaliana”will be more suitable for this text file.

Round 2
Reviewer 2 Report
The manuscript can be accepted.